# EFHC1, implicated in juvenile myoclonic epilepsy, functions at the cilium and synapse to modulate dopamine signaling

Catrina M Loucks[1,2†], Kwangjin Park[1,2†], Denise S Walker[3], Andrea H McEwan[4], Tiffany A Timbers[1,2], Evan L Ardiel[4], Laura J Grundy[3], Chunmei Li[1,2], Jacque-Lynne Johnson[1,2], Julie Kennedy[5], Oliver E Blacque[5], William Schafer[3], Catharine H Rankin[4,6], Michel R Leroux[1,2]*

[1]Department of Molecular Biology and Biochemistry, Simon Fraser University, Burnaby, Canada; [2]Centre for Cell Biology, Development, and Disease, Simon Fraser University, Burnaby, Canada; [3]Neurobiology Division, MRC Laboratory of Molecular Biology, Cambridge, United Kingdom; [4]Djavad Mowfaghian Centre for Brain Health, University of British Columbia, Vancouver, Canada; [5]School of Biomolecular and Biomedical Science, University College Dublin, Dublin, Ireland; [6]Department of Psychology, University of British Columbia, Vancouver, Canada

*For correspondence: leroux@sfu.ca

[†]These authors contributed equally to this work

Competing interests: The authors declare that no competing interests exist.

**Abstract** Neurons throughout the mammalian brain possess non-motile cilia, organelles with varied functions in sensory physiology and cellular signaling. Yet, the roles of cilia in these neurons are poorly understood. To shed light into their functions, we studied EFHC1, an evolutionarily conserved protein required for motile cilia function and linked to a common form of inherited epilepsy in humans, juvenile myoclonic epilepsy (JME). We demonstrate that *C. elegans* EFHC-1 functions within specialized non-motile mechanosensory cilia, where it regulates neuronal activation and dopamine signaling. EFHC-1 also localizes at the synapse, where it further modulates dopamine signaling in cooperation with the orthologue of an R-type voltage-gated calcium channel. Our findings unveil a previously undescribed dual-regulation of neuronal excitability at sites of neuronal sensory input (cilium) and neuronal output (synapse). Such a distributed regulatory mechanism may be essential for establishing neuronal activation thresholds under physiological conditions, and when impaired, may represent a novel pathomechanism for epilepsy.
DOI: https://doi.org/10.7554/eLife.37271.001

## Introduction

Juvenile myoclonic epilepsy (JME) is the most common form of idiopathic epilepsy in humans, making up 10–30% of cases (*Delgado-Escueta, 1984*). Most genes implicated in idiopathic epilepsies encode ion channels with predictable consequences on neuronal excitability, yet disruption of EFHC1, a non-ion channel protein, has been reported as the most frequent cause of JME (*Delgado-Escueta, 2007*). While several neuronal roles have been proposed for EFHC1 (*de Nijs et al., 2012*; *de Nijs et al., 2006*; *de Nijs et al., 2009*; *Rossetto et al., 2011*; *Suzuki et al., 2004*), it remains unclear how mutations in *EFHC1* lead to epilepsy. Interestingly, EFHC1 is also specifically associated with motile cilia that project from specialized cells to enable fluid flow (*Conte et al., 2009*; *Suzuki et al., 2008*; *Suzuki et al., 2009*), but a role for EFHC1 in non-motile cilia that emanate from most cell types to allow for sensory and signaling functions largely remains unexplored (*Zhao et al., 2016*).

In the human brain, there are both motile cilia, which enable cerebrospinal fluid flow, and non-motile cilia, which project from the cell bodies of all or most neurons (*Bishop et al., 2007*). Although

EFHC1 dysfunction impairs motility of ependymal cilia in mice, the accompanying ventricle enlargement does not correlate with epilepsy (*Suzuki et al., 2009*). Furthermore, although several neuronal functions have been attributed to EFHC1, including regulation of ion channels, apoptosis, cell division, neuronal migration, neurite architecture and neurotransmitter release (*de Nijs et al., 2012*; *de Nijs et al., 2006*; *de Nijs et al., 2009*; *Rossetto et al., 2011*; *Suzuki et al., 2004*), a possible role for EFHC1 in non-motile cilia of neuronal cells has not been explored. As a core component of the protofilament ribbon structure of motile cilia thought to dictate the ordered attachment/arrangement of proteins required for motility (*Ikeda et al., 2003*; *Linck et al., 2014*), EFHC1 may play a similar role in neuronal cilia to anchor/regulate signaling molecules and modulate neuronal excitability.

Here we demonstrate that rather than being a core component of ciliary motility-associated machinery, the *C. elegans* orthologue of EFHC-1 is required for mechanosensation in a class of non-motile cilia, where it regulates neuronal activation and dopamine signaling. Interestingly, EFHC-1-mediated signaling also occurs at the synapse in cooperation with a known EFHC1-interaction partner, an R-type voltage-gated calcium channel. Our work highlights the importance of exaptation (functional adaptation) of a cilium motility protein in non-motile sensory cilia. Moreover, our findings reveal a previously undescribed dual-regulation of neuronal excitability at the site of sensory neuron input (cilium) and sensory neuron output (synapse) and suggest an important correspondance between dopamine neurotransmission and epilepsy.

## Results

### EFHC-1 localizes to cilia and synapses of mechanosensory dopaminergic neurons and to the distal regions of male-specific dopaminergic ray neurons

To isolate possible non-motile ciliary functions of EFHC1, we took advantage of *Caenorhabditis elegans*, a nematode which only has non-motile sensory cilia. We generated a GFP reporter of *C. elegans* EFHC-1 driven by its own promoter, and found that the *efhc-1::gfp* fusion construct is specifically expressed in a small subset of ciliated mechanosensory neurons: the dopaminergic CEP, ADE and PDE neurons, and the glutamatergic OLQ neurons (*Figure 1A*). Both classes of neurons terminate in cilia embedded in the cuticle, ideally positioned to mediate mechanosensation at various locations along the body: four CEP neurons paired with four OLQ neurons project cilia in the nose, two ADE neurons position cilia in the anterior body, and two PDE neurons localize cilia to the posterior body (*Inglis et al., 2007*; *Sulston et al., 1975*; *White et al., 1986*) (*Figure 1A*). Additionally, EFHC-1::GFP is found in the distal regions of sex-specific dopaminergic ray neurons in the male tail (*Figure 1B*, *Figure 1—figure supplement 1*). Importantly, these expression patterns were confirmed by tagging endogenous EFHC-1 with GFP using the CRISPR/Cas9 technique (*Figure 1—figure supplement 2A–B*).

The ciliated dopaminergic neurons mechanically sense bacteria to modulate behaviors based on food availability (*Ezcurra et al., 2011*; *Kindt et al., 2007a*; *Sanyal et al., 2004*; *Sawin et al., 2000*), while OLQ neurons sense light touch to the nose and control the rate and bending angle of foraging (side-to-side head movements) (*Hart et al., 1995*; *Kaplan and Horvitz, 1993*). In these ciliated neurons, the generated EFHC-1::GFP fusion protein is enriched within cilia, alongside the intraflagellar transport protein (IFT) XBX-1 (DLIC orthologue) which marks ciliary basal bodies and axonemes (*Figure 1C*). Interestingly, while endogenously GFP-tagged EFHC-1 mostly accumulates in both OLQ and ADE cilia, it is weakly present in CEP cilia (*Figure 1—figure supplement 2A*), possibly reflecting a lower abundance and/or a tight incorporation with the microtubules at regular intervals within the CEP cilia that limits/masks the fluorescence associated with endogenously GFP-tagged EFHC-1. Ciliary localization of EFHC-1 is further supported by the ability of a truncated protein (C-terminal fragment containing the second of two evolutionarily conserved DM10 domains of unknown function) to specifically localize to OLQ, CEP, ADE and PDE cilia in a manner indistinguishable from the full-length protein (*Figure 1D* and *Figure 1—figure supplement 3*). This is in contrast to a second truncated protein (N-terminal fragment containing the first DM10 domain) that displays non-specific localization throughout all neuronal processes of glutamatergic (OLQ) and dopaminergic (CEP, ADE and PDE) neurons (*Figure 1D* and *Figure 1—figure supplement 3*).

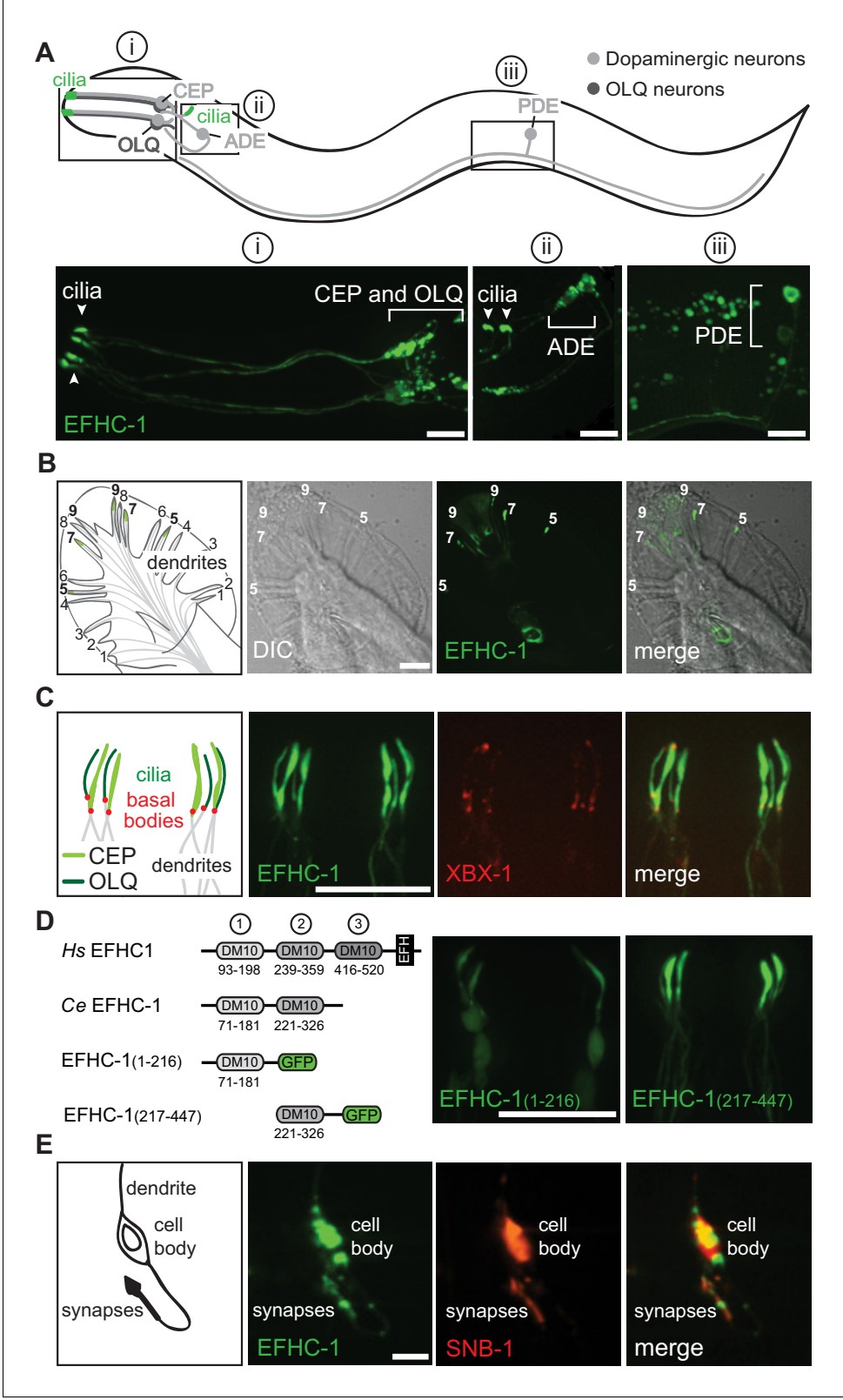

**Figure 1.** EFHC-1 localizes to cilia and synapses of mechanosensory dopaminergic neurons and to the distal regions of male-specific dopaminergic ray neurons. (**A**) EFHC-1::GFP protein marker is expressed in all classes of dopaminergic neurons shared by both hermaphrodites and males (CEP, ADE and PDE), as well as in the mechanosensory OLQ neurons. Localization is enriched in cilia (arrowheads). Scale bar, 10 μm. (**B**) EFHC-1::GFP is

*Figure 1 continued on next page*

*Figure 1 continued*

also expressed in dopaminergic ray RnA neurons in the male tail (rays 5, 7 and 9) distal to dendrites. Scale bar, 10 μm. (C) In CEP and OLQ cilia, EFHC-1::GFP localization surrounds the axoneme marked by the intraflagellar transport protein XBX-1::tdTomato. Scale bar, 10 μm. (D) Human (*Hs*) EFHC1 is composed of three DM10 domains of unknown function and a C-terminal EF-hand (EFH) domain that is thought to bind calcium. In *C. elegans* (*Ce*), only the first two DM10 domains are conserved. A truncation protein harboring the first DM10 domain shows non-specific localization, while a truncation protein containing only the second DM10 domain shows specific ciliary localization in OLQ and CEP neurons. Scale bar, 10 μm. (E) EFHC-1::GFP also localizes to synaptic regions in CEP neurons, partially overlapping with the synaptic protein, SNB-1::tdTomato. Scale bar, 10 μm.
DOI: https://doi.org/10.7554/eLife.37271.002
The following figure supplements are available for figure 1:

**Figure supplement 1.** The EFHC-1::GFP fusion protein localizes to the dopaminergic ray neurons in the male tail.
DOI: https://doi.org/10.7554/eLife.37271.003
**Figure supplement 2.** The expression patterns of EFHC-1 in an *efhc-1::gfp* knock-in strain mirror those seen for the EFHC-1::GFP fusion protein.
DOI: https://doi.org/10.7554/eLife.37271.004
**Figure supplement 3.** An EFHC-1 truncation with only the first DM10 domain shows non-specific localization throughout OLQ and dopaminergic neurons, while a protein containing only the second DM10 domain shows specific ciliary localization within these neurons.
DOI: https://doi.org/10.7554/eLife.37271.005
**Figure supplement 4.** Localization of EFHC-1 within cilia is largely static and independent of typical intraflagellar transport (IFT) mechanisms within cilia, the dendritic/ciliary transport protein, UNC-101, and the ciliary transcription factor, DAF-19.
DOI: https://doi.org/10.7554/eLife.37271.006
**Figure supplement 5.** Preynaptic localization of synaptic vesicle protein RAB-3 is dependent on UNC-104, while EFHC-1 and active zone protein ELKS-1 are independent of UNC-104.
DOI: https://doi.org/10.7554/eLife.37271.007

Given that EFHC-1 is a core component of the protofilament ribbon structure of motile cilia that closely associates with microtubules and is ideally positioned to anchor proteins at specific locations along cilia, we hypothesized that ciliary localization of EFHC-1 is largely static. Consistent with this, when either the distal or middle portions of CEP and OLQ cilia are photobleached, the EFHC-1::GFP signal does not rapidly recover within 30 s, as would be expected for proteins dependent on diffusion to maintain their proper ciliary localizations, such as ARL-13 (*Cevik et al., 2013*) (*Figure 1—figure supplement 4A–B*). Moreover, kymograph analyses suggest that EFHC-1 proteins present in CEP and OLQ cilia are not associated with IFT (*Figure 1—figure supplement 4C*). EFHC-1 localization is also not dependent on UNC-101, a protein required for dendritic transport and proper ciliary localization of a set of ciliary proteins (*Dwyer et al., 2001*) (*Figure 1—figure supplement 4D*). For example, ODR-10 expressed in AWB olfactory neurons of wild-type animals predominantly accumulates in cilia, but in *unc-101* mutant animals it no longer concentrates inside cilia and diffuses to dendrites, cell bodies, and axons (*Figure 1—figure supplement 4D*). Interestingly, ciliary expression of EFHC-1 is not dependent on DAF-19 (*Figure 1—figure supplement 4E*), a transcription factor required for the expression of many ciliary genes, again suggesting the unique nature of this ciliary protein, typically associated with motility, which has been retained in *C. elegans* to facilitate ciliary mechanosensation.

In addition to EFHC-1's ciliary localization, we find that the EFHC-1::GFP fusion protein is present at presynaptic regions in dopaminergic neurons, partially overlapping with the synaptic vesicle protein, synaptobrevin (SNB-1) (*Figure 1E*). By tagging endogenous EFHC-1 with GFP, we confirm that EFHC-1 localizes to these presynaptic regions, where it partially overlaps with a second synaptic vesicle protein (RAB-3) (*Figure 1—figure supplement 5A*). More specifically, EFHC-1 is found to partially colocalize with the active zone protein (ELKS-1) (*Figure 1—figure supplement 5B–C*), where it is ideally positioned to regulate synaptic release in *C. elegans*. Since the presynaptic localization of synaptic vesicle proteins such as RAB-3, but not some active zone proteins like ELKS-1, depends on the kinesin-3 motor, UNC-104 (*Deken et al., 2005*; *Nonet et al., 1997*), we examined whether EFHC-1 also requires UNC-104 for its synaptic localization in CEP neurons. While RAB-3 localization is abrogated in *unc-104* mutants, accumulating in cell bodies and diffusing to dendrites/axons of

CEP neurons, UNC-104 is dispensable for both ELKS-1 and EFHC-1 localization (and consequently their partial colocalization pattern) to presynaptic regions (*Figure 1—figure supplement 5A–C*). The presence of EFHC-1 at both the cilium and presynaptic regions of dopaminergic neurons suggests that it may play important roles in both sensation and regulation of neuronal output.

## EFHC-1 influences habituation to a mechanical stimulus by modulating dopamine signaling

We first investigated a possible role for EFHC-1 in dopamine signaling. In the presence of food, ciliary TRP-4 channels (orthologous to TRPN/NOMPC) are mechanically activated, resulting in the release of dopamine that modulates several behaviors through dopamine receptors (DOP-1, DOP-3 and DOP-4) on various postsynaptic/target cells (*Chase et al., 2004*; *Ezcurra et al., 2011*; *Sanyal et al., 2004*) (*Figure 2A*). One such behavior is the basal slowing response, where wild-type animals slow in the presence of food. Disrupting tyrosine hydroxylase, CAT-2, a rate-limiting enzyme for dopamine synthesis, impairs dopamine signaling and prevents slowing in the presence of food (*Sawin et al., 2000*) (*Figure 2A*). In contrast, loss-of-function *efhc-1(gk424336)* mutants are able to slow in the presence of food (*Figure 2—figure supplement 1A*), suggesting that they can release dopamine in the presence of food. Given that EFHC-1 also localizes to the distal regions of male dopaminergic neurons that are required for the coordination of locomotory events that enable male mating behaviours (*Correa et al., 2012*), we assessed if *efhc-1* mutants are able to release dopamine to facilitate male mating. As expected, while *cat-2* mutant males show reduced mating potency due to impaired dopamine signaling, *efhc-1* mutant males show mating potency similar to wild-type animals (*Figure 2—figure supplement 1B*). Together, these results show that *efhc-1* mutants are able to release dopamine in appropriate behavioural contexts.

To investigate the possibility that dopamine signaling may be increased in *efhc-1* mutants, we assessed their behavior in tap habituation, a form of non-associative learning and memory (*Rankin et al., 1990*) modulated by dopamine (*Kindt et al., 2007a*; *Sanyal et al., 2004*). When animals experience a non-localized mechanical stimulus (tap), the touch receptor neurons are stimulated, and worms execute an escape response most often seen as reversals (*Rankin et al., 1990*) (*Figure 2—figure supplement 2A*). The proportion of animals responding with a reversal decreases with repeated taps (*Rankin et al., 1990*), and dopamine modulates this habituation (*Kindt et al., 2007a*; *Sanyal et al., 2004*). Decreased dopamine signaling in *cat-2* mutants results in fast habituation, while increased dopamine signaling in mutants that lack the dopamine re-uptake transporter DAT-1 results in slow habituation (*Kindt et al., 2007a*; *Sanyal et al., 2004*) (*Figure 2B*). Habituation in *efhc-1* mutants closely resembles that of *dat-1* mutants, suggesting that abrogating EFHC-1 increases dopamine signaling (*Figure 2B*). A second *efhc-1* allele (*tm6235*) lacking most of the gene also presents a slow habituation phenotype (*Figure 2C*), confirming that slow habituation is a result of EFHC-1 dysfunction. Furthermore, introducing the *cat-2* mutation into the *efhc-1* genetic background abolishes the slow habituation phenotype, resulting in fast habituation indistinguishable from *cat-2* (*Figure 2D*). This indicates that the slow habituation phenotype in *efhc-1* mutant animals requires (stems from) increased dopamine signaling.

To further investigate the dopamine signaling defect in *efhc-1* mutants, we analysed their behavior in the absence of food, and found that they again show slow habituation compared to wild-type animals (*Figure 2—figure supplement 2B*). Interestingly, *efhc-1* mutants off food mimic the habituation phenotype of wild-type animals on food, while *dat-1* mutants habituate slightly slower than *efhc-1* mutants off food *Figure 2—figure supplement 2C*). This suggests that although both mutants increase dopamine signaling, they likely do so in different ways, and to different extents. We further confirm this possibility by exploiting the dopamine-dependent swimming-induced paralysis (SWIP) phenotype seen in *dat-1* mutants, which is attributed to excessive dopamine signaling (*McDonald et al., 2007*) (*Figure 2A*). Whereas most wild-type animals continue swimming 10 min after being submersed in water, most *dat-1* animals are paralysed (*Figure 2E*), exhibiting an average thrashing frequency consistently lower than wild-type animals (*Figure 2—figure supplement 2D*). In this assay, both *efhc-1* alleles show intermediate levels of paralysis at 10 min compared to wild-type and *dat-1* mutants (*Figure 2E*), and corresponding intermediate thrashing frequencies (*Figure 2—figure supplement 2D*). Together, our data are consistent with *efhc-1* mutants having increased dopamine signaling compared to wild-type animals, but less than *dat-1* mutants.

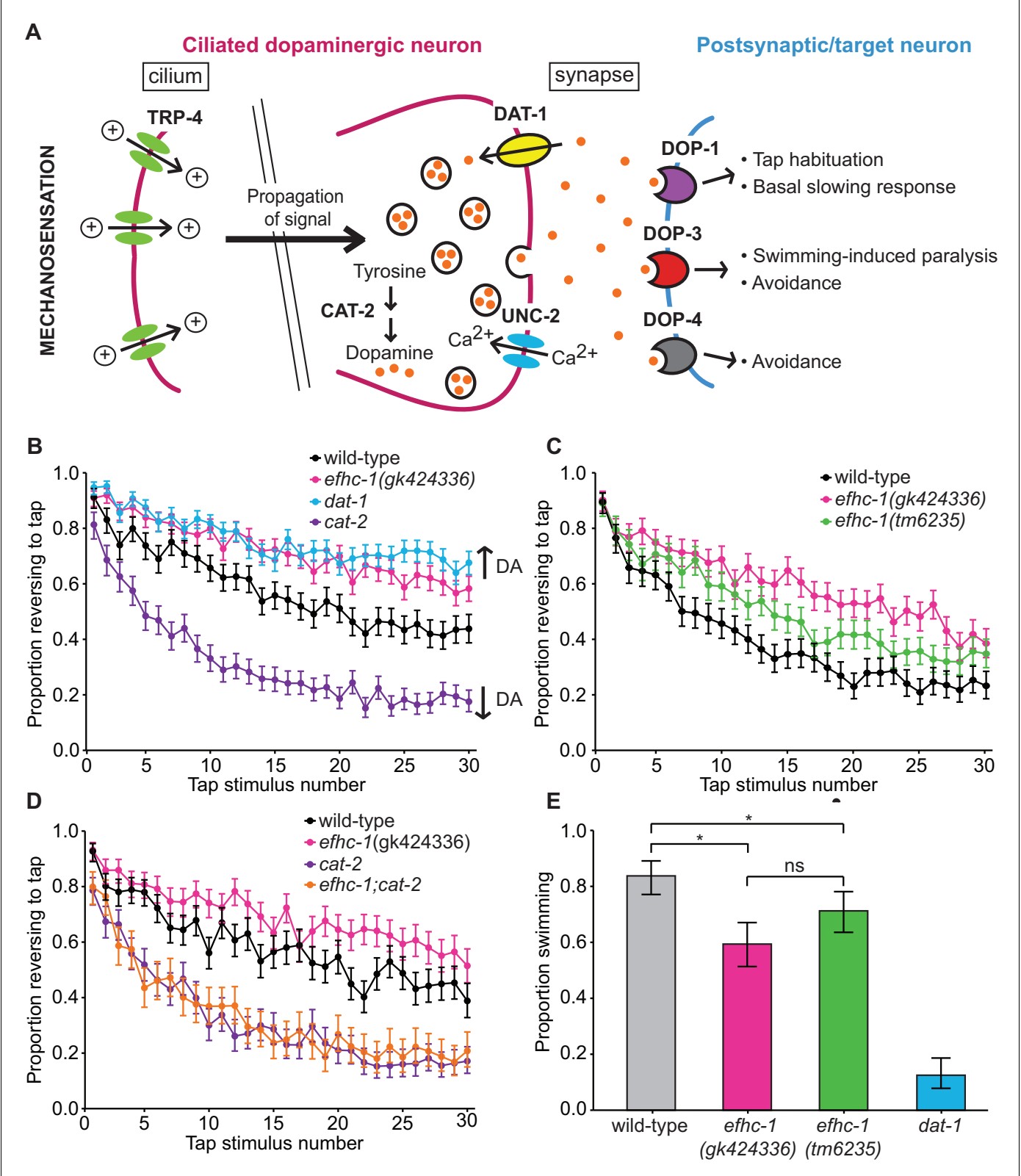

**Figure 2.** EFHC-1 influences habituation to a mechanical stimulus by modulating dopamine signaling. (**A**) The molecular components of dopamine signaling in *C. elegans*. Food (bacteria) mechanically stimulates the ciliated dopaminergic neurons (dark pink), resulting in the release of dopamine that modulates several behaviors through dopamine receptors on various postsynaptic/target cells (blue). (**B**) In the presence of food, *efhc-1* mutants habituate slower than wild-type animals, and similar to *dat-1* mutants that have increased dopamine signaling (DA: dopamine). At each tap, the points

*Figure 2 continued on next page*

*Figure 2 continued*

represent the number of worms reversing over the total number of worms tracked per strain, and the error bars represent the binomial confidence intervals. Binomial logistic regression followed by Tukey's HSD (honest significant difference) test was used to determine significance of the habituated level (proportion reversing at tap 30) for each pair of strains. All strains are statistically different from each other (p < 0.001). (C) A second allele of *efhc-1, tm6235,* also leads to a slow habituation phenotype that is statistically different from wild-type (p < 0.001). All strains are statistically different from each other (p < 0.001). (D) The habituation phenotype seen in *efhc-1* mutants is abolished in *efhc-1;cat-2* double mutants, suggesting that the slow habituation phenotype of *efhc-1* mutants depends on dopamine. All strains are statistically different from each other (p < 0.001) except for *cat-2* and *efhc-1;cat-2* (p = 0.69). (E) *efhc-1* mutants also show defects in the dopamine-dependent swimming-induced paralysis phenotype, with an intermediate level of paralysis after 10 min compared to wild-type and *dat-1* mutants. Each bar represents the number of worms swimming after 10 min over the total number of worms per strain, and the error bars represent the binomial confidence intervals (n = 160 for each strain). Binomial logistic regression followed by Tukey's HSD (honest significant difference) test was used to determine significance for each pair of strains. All strains are statistically different from each other except for the two *efhc-1* strains (ns = not significant, p = 0.115). For sample sizes (n) for tap habituation experiments and complete statistical comparisons (p values) see *Supplementary files 1* and *2*, respectively.

DOI: https://doi.org/10.7554/eLife.37271.008

The following figure supplements are available for figure 2:

**Figure supplement 1.** *efhc-1* mutants are able to release dopamine in the presence of appropriate behavioral stimuli.

DOI: https://doi.org/10.7554/eLife.37271.009

**Figure supplement 2.** *efhc-1* mutants show behavioral phenotypes consistent with increased dopamine signaling.

DOI: https://doi.org/10.7554/eLife.37271.010

## EFHC-1, similar to TRP-4, is required for mechanosensation in non-motile CEP cilia

A possible explanation for the increased dopamine signaling in *efhc-1* mutants is increased ciliary activation, even in the absence of food. To directly interrogate mechanosensory activation in *efhc-1* mutants, we used a genetically-encoded calcium indicator (cameleon) specifically expressed in dopaminergic neurons (*Kindt et al., 2007a*). Two different mechanical stimuli to the nose were applied using a glass probe: a gentle stimulus consisting of vibrations to the nose for 3.7 s, expected to mimic the presence of food (buzz), and a stronger stimulus of a firm press to the nose for 1 s (press) (*Kindt et al., 2007a*). Wild-type animals show a calcium response to each stimulus ~50% of the time, as measured by a change in the YFP/CFP ratio (*Figure 3A*). In contrast, *efhc-1(gk424336)* mutants rarely respond upon gentle stimulation but respond at wild-type levels to the stronger stimulation (*Figure 3A*). Interestingly, when *efhc-1* mutants do exhibit calcium responses to either stimulus, they are of a comparable amplitude and shape as wild-type responses (*Figure 3B* and *Figure 3—figure supplement 1*). This behavior is remarkably similar to that seen in animals lacking the cilium-localized mechanosensitive TRP-4 channel, which is required for food sensation (*Kang et al., 2010*; *Kindt et al., 2007a*). This suggests that like TRP-4, EFHC-1 has a specific role in mediating the likelihood of mechanosensory transduction upon gentle stimulation, as seen when an animal encounters food. In contrast, mechanosensory transduction in non-dopaminergic OLQ neurons is not impaired in *efhc-1* mutants and OLQ-mediated foraging behaviors are similarly unaffected (*Figure 3—figure supplement 2A–E*). This implies a specific mechanosensory function for EFHC-1 in CEP dopaminergic neurons.

Given the hypothesis that EFHC-1 helps to anchor, position, and/or regulate ciliary proteins along the ciliary axoneme in motile cilia, it is possible that it has been adapted in non-motile cilia to anchor sensory proteins such as TRP-4. We first confirmed by transmission electron microscopy (TEM) that 'conventional' cilia not harboring EFHC-1::GFP (as determined by our localization analyses) are structurally normal in *efhc-1* mutants, displaying nine outer doublet microtubules that become singlets in more distal regions (*Figure 3—figure supplement 3*). Notably, EFHC-1-containing CEP and OLQ cilia—which have supernumerary singlet microtubules interspersed with tubule-associated material or unique arrangement of doublets in their distal regions, respectively— also appear structurally normal in *efhc-1* mutants (*Figure 3C* and *Figure 3—figure supplement 2F*). Hence, EFHC-1 does not play an essential role in specifying, or maintaining, non-motile cilia ultrastructure in this metazoan. Furthermore, TRP-4::YFP localization along the axoneme of CEP cilia is retained in *efhc-1* mutants (*Figure 3D*), suggesting that in the absence of EFHC-1, ciliary localization of TRP-4 still occurs, but its function might be perturbed. For example, EFHC-1 may be required for regulating TRP-4 activity, or anchoring/regulating other signaling molecules required for mechanosensory transduction.

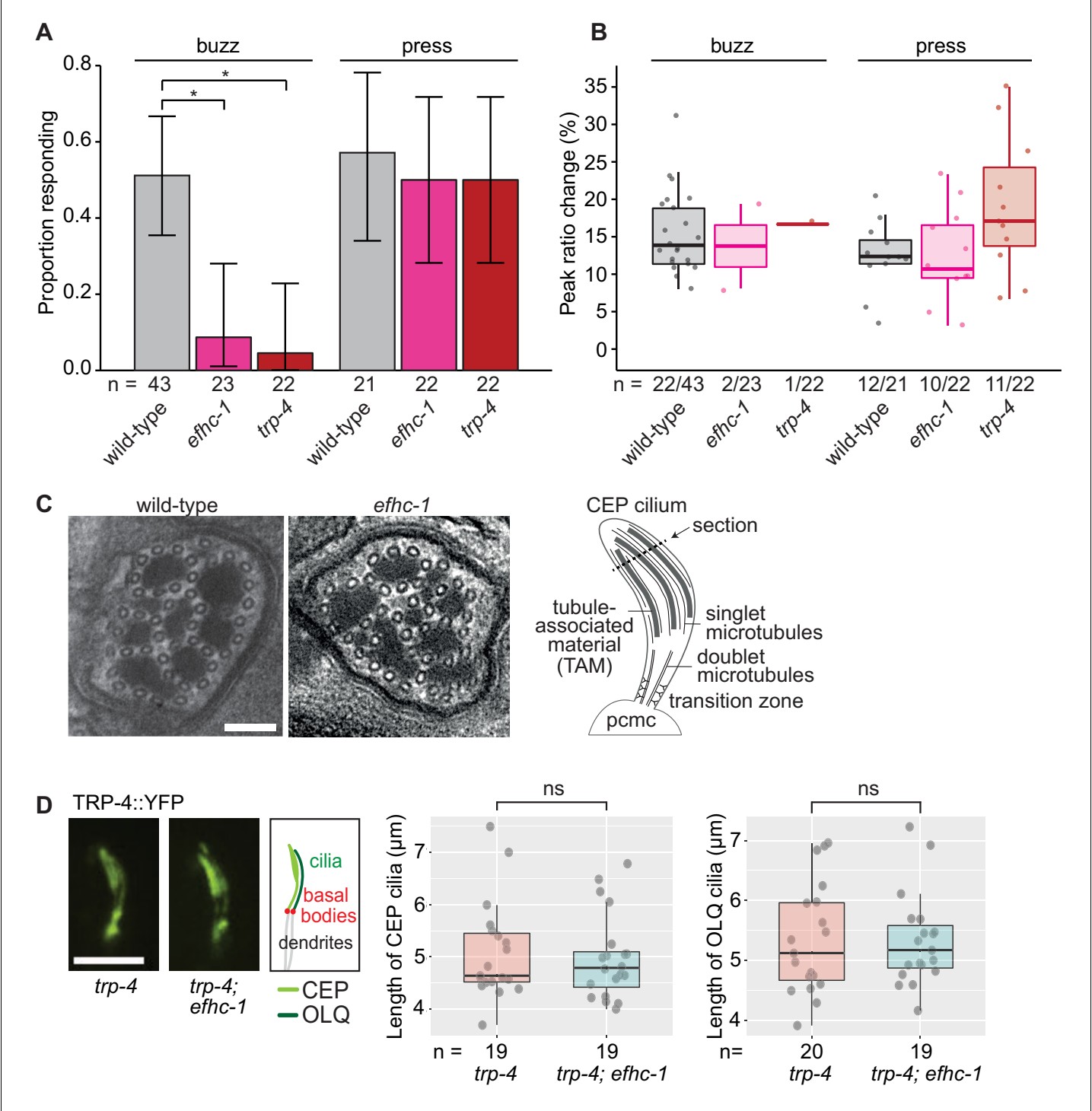

**Figure 3.** EFHC-1, similar to TRP-4, is required for mechanosensation in non-motile CEP cilia. (**A**) Loss of EFHC-1 or TRP-4 impairs activation of CEP neurons upon gentle mechanical stimulation (buzz), resulting in fewer animals responding to the stimulus compared to wild-type animals. In contrast, *efhc-1* or *trp-4* mutants can respond to stronger stimulation (press). In this panel, n represents the total number of animals assessed. Each bar represents the number of worms responding to mechanical stimulation over the total number of worms assessed per strain, and the error bars represent the binomial confidence intervals. Binomial logistic regression followed by Tukey's HSD (honest significant difference) test was used to determine significance for each pair of strains. Differences between all strains subjected to the press stimulus are not significant, but differences between wild-type and *efhc-1* (p < 0.01) or *trp-4* (p < 0.05) mutants subjected to the buzz stimulus are significant. (**B**) The peak ratio change (%), representing the amplitude of the neuronal response, is similar between strains for animals responding to each stimulus. In this panel, n represents the

*Figure 3 continued on next page*

*Figure 3 continued*
number of animals responding to the stimulus over the total number of animals assessed. (C) The axonemal structure of CEP neurons by TEM cross-section (refer to schematic) is not abrogated in *efhc-1* mutants. Scale bars, 100 nm. (D) The localization of the mechanosensory channel, TRP-4, which is comparable to that of EFHC-1, is unaffected in *efhc-1* mutants. Lengths of CEP and OLQ cilia measured in *trp-4* and *trp-4;efhc-1* mutants are indistinguishable. Each dot represents one cilium. Kruskal-Wallis rank sum test (CEP cilia lengths) and Tukey's Honest Significant Differences (OLQ cilia lengths) were used for calculating the statistical significance. ns, not significant. Scale bar, 3 µm. For complete statistical comparisons (p values) see *Supplementary file 2*.
DOI: https://doi.org/10.7554/eLife.37271.011
The following figure supplements are available for figure 3:

**Figure supplement 1.** Animals that respond to mechanical stimuli in the CEP neurons show similar patterns of calcium changes.
DOI: https://doi.org/10.7554/eLife.37271.012
**Figure supplement 2.** *efhc-1* mutants show no defects in OLQ-related mechanosensory or behavioral responses, or OLQ ciliary structure.
DOI: https://doi.org/10.7554/eLife.37271.013
**Figure supplement 3.** General ciliary structure is retained in *efhc-1* mutants.
DOI: https://doi.org/10.7554/eLife.37271.014

## EFHC-1-mediated regulation of dopamine signaling occurs, in part, downstream of cilia

Given that mechanical activation by food is required for dopamine release and that EFHC-1 is required for efficient mechanosensation, *efhc-1* mutants were expected to show phenotypes consistent with decreased dopamine signaling. However, the increased dopamine signaling in *efhc-1* mutants suggests that EFHC-1 may regulate dopamine release at the synapse, independent of its role in ciliary-based signal transduction. Specifically, while loss of EFHC-1 function at the cilium reduces the likelihood of mechanical activation by food, its loss at the synapse may stimulate the release of dopamine in the absence of mechanosensory stimuli and/or amplify infrequent mechanosensory stimuli. We first confirmed that synaptic regions are intact in *efhc-1* mutants (*Figure 4—figure supplement 1*). Next, we asked if the fast habituation seen in *trp-4* mutants, due to an inability to mechanically sense bacteria at the cilium and subsequently release dopamine (*Figure 2A*) (*Kindt et al., 2007a*), is affected by loss of EFHC-1 function. Interestingly, *efhc-1;trp-4* double mutants show slower habituation than *trp-4* mutants (*Figure 4A*), consistent with a rise in dopamine signaling independent of ciliary TRP-4-associated mechanosensation. In other words, loss of EFHC-1 function at the synapse may inappropriately stimulate dopamine release in the absence of TRP-4-mediated mechanosensation.

To confirm that EFHC-1 functions downstream of cilia, likely at the synapse, to modulate dopamine signaling, *efhc-1* was combined with a *daf-19* mutant lacking cilia in many sensory neurons, including dopaminergic neurons (*daf-19* encodes a transcription factor required for the expression of many ciliary genes, but not *efhc-1* - see *Figure 1—figure supplement 4E*). Interestingly, *daf-19* mutants respond weakly to tap stimuli and habituate rapidly, likely due to loss or misregulation of many cilium-dependent sensory inputs required to fine-tune mechanosensation and habituation; notably, this phenotype is similar to the loss of TRP-4, which requires cilia to function (*Figure 4B*). In contrast, the *efhc-1;daf-19* double mutants show responsiveness and habituation phenotypes that are intermediate between the *daf-19* and *efhc-1* single mutants (*Figure 4B*), again suggesting that the loss of EFHC-1 results in increased dopamine signaling downstream of ciliary function. These data, together with the localization of EFHC-1 at synaptic regions, are consistent with EFHC-1 preventing spontaneous release of dopamine at synapses in the absence of mechanical stimulation by food.

The release of dopamine at the synapse requires the activation of voltage-gated calcium channels at presynaptic termini, which allows for calcium influx and subsequent dopamine-bearing vesicle fusion (*Figure 2A*). In humans, EFHC1 physically interacts with and modulates the alpha subunit of the R-type voltage-gated calcium channel (*Suzuki et al., 2004*), which is most closely related to UNC-2 in *C. elegans*. We therefore hypothesized that EFHC-1 prevents spontaneous activity of UNC-2 in *C. elegans* dopaminergic neurons, such that loss of *efhc-1* would stimulate UNC-2 activity even in the absence of mechanosensation at the cilium and result in increased dopamine release. Since *unc-2* mutants are uncoordinated and cannot be tested for habituation defects, we took advantage of a gain-of-function (gf) allele that shows slow habituation compared to both wild-type

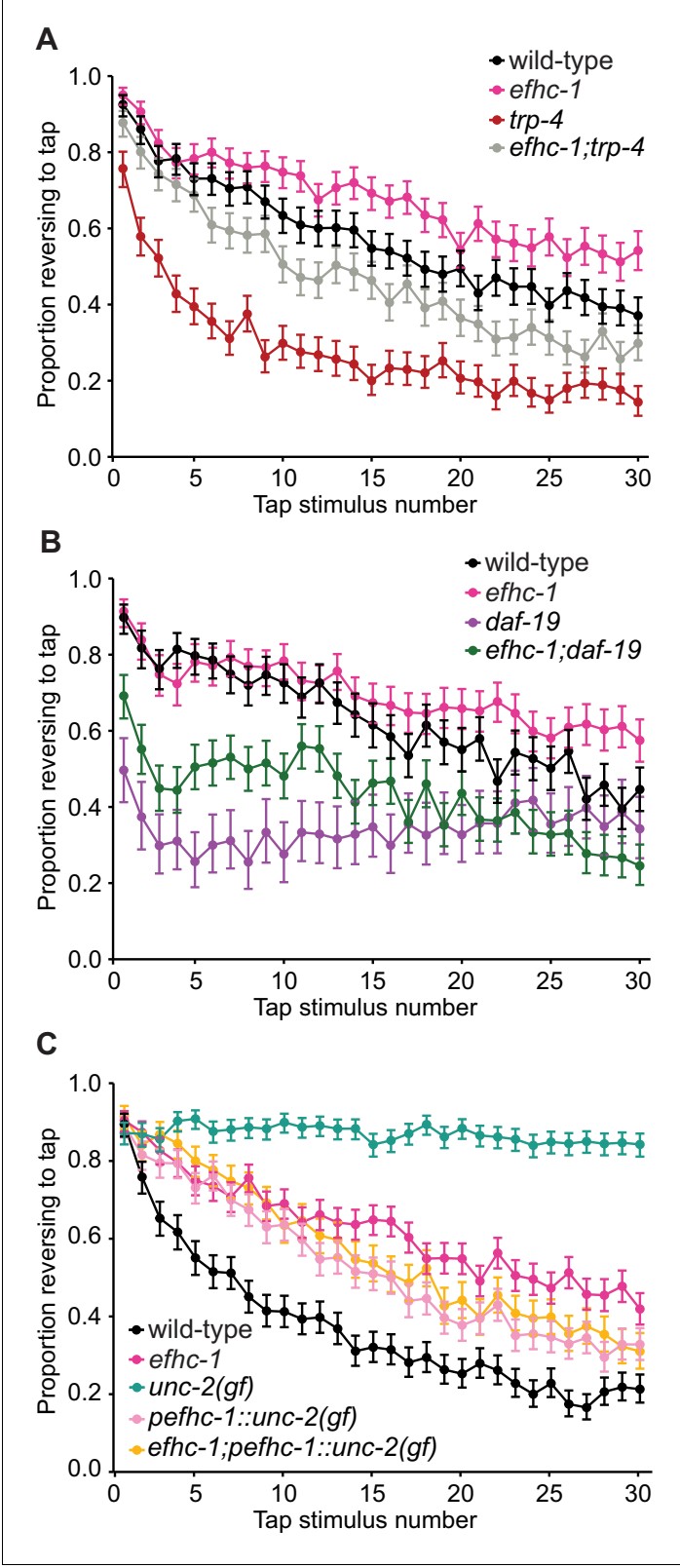

**Figure 4.** EFHC-1-mediated regulation of dopamine signaling occurs, in part, downstream of cilia. (**A**) Double mutants of *efhc-1* and *trp-4* show intermediate habituation compared to the *trp-4* and *efhc-1* single mutants. At each tap, the points represent the number of worms reversing over the total number of worms tracked per strain, and the error bars represent the binomial confidence intervals. Binomial logistic regression followed by Tukey's

*Figure 4 continued*

HSD (honest significant difference) test was used to determine significance of the habituated level (proportion reversing at tap 30) for each pair of strains. All strains are statistically different from each other (p < 0.001). (B) *efhc-1* mutants with absent cilia due to a *daf-19* mutation show intermediate habituation compared to *daf-19* and *efhc-1* single mutants. All strains are statistically different from each other (p < 0.001). (C) An *unc-2(gain-of-function)* mutation driven in EFHC-1-expressing cells shows slow habituation compared to wild-type animals, but faster habituation compared to animals with the *unc-2(gain-of-function)* mutation expressed throughout. Furthermore, *efhc-1* mutants with the addition of an *unc-2(gain-of-function)* mutation driven in EFHC-1-expressing cells does not exacerbate the habituation phenotype. All strains are statistically different from each other (p < 0.001). For sample sizes (n) for tap habituation experiments and complete statistical comparisons (p values) see *Supplementary files 1* and *2*, respectively.

DOI: https://doi.org/10.7554/eLife.37271.015

The following figure supplement is available for figure 4:

**Figure supplement 1.** Synaptic structure in *efhc-1* mutants is indistinguishable from wild-type.

DOI: https://doi.org/10.7554/eLife.37271.016

animals and *efhc-1* mutants (*Figure 4C*). Also, because *unc-2* expression is found in many neurons apart from the dopaminergic neurons that may modulate tap habituation, a construct was made with UNC-2(gf) under the control of the *efhc-1* promoter. Remarkably, expressing overactive UNC-2 specifically in *efhc-1*-expressing cells results in a slow habituation phenotype similar (but not identical) to that seen in *efhc-1* mutants; crossing the UNC-2(gf) construct into *efhc-1* mutants did not result in slower habituation, but instead leads to habituation similar to that seen in animals with UNC-2(gf) specifically in *efhc-1*-expressing cells (*Figure 4C*). Similarly, these trends were replicated in SWIP assays (*Figure 2—figure supplement 2E*). Our findings therefore provide evidence that independent of its sensory/activation role at the cilium, EFHC-1 may negatively regulate UNC-2, such that loss of EFHC-1 results in maximum activation of the voltage-gated channel at the synapse.

## Discussion

Although the cilia of multicellular organisms are typically classified as either motile or non-motile, unicellular organisms possess one or more cilia responsible for both producing movement and responding to mechanosensory cues to modulate movement (*Figure 5*). Despite an established role for EFHC1 in ciliary motility (*Conte et al., 2009*; *Suzuki et al., 2008*; *Suzuki et al., 2009*), its disruption has no effect on ciliary structure (*Suzuki et al., 2009*), raising the possibility that it plays a role in regulating motility—for example in the mechanosensory feedback required to support normal ciliary motility. Our work in a representative eumetazoan (*C. elegans*) that lacks motile cilia supports this idea, where EFHC-1 has been adapted to function in a specific class of non-motile ciliated mechanosensory cells, the dopaminergic neurons (see model in *Figure 5*), where it is required for mechanosensation and proper dopamine signaling, yet dispensable for non-motile ciliary structure.

Our findings suggest that EFHC1, by being an integral part of the ciliary axoneme and by potentially modulating TRP (or other types of) mechanosensitive calcium channels (*Katano et al., 2012*) may have adapted its mechanosensation role from motile cilia (*e.g.*, for environmental sensing and/ or internal sensation and regulation of motile cilium movement) to a strictly sensory role in non-motile cilia. We also find that in *C. elegans*, EFHC-1 functions at the synapses of dopaminergic neurons (*Figure 5*). Importantly, this role is also likely related to the modulation of a calcium channel, namely UNC-2, that localizes to presynaptic active zones of sensory neurons (*Saheki and Bargmann, 2009*), where it may play an important role in evoked neurotransmitter release (*Mathews et al., 2003*; *Richmond et al., 2001*; *Schafer and Kenyon, 1995*). Together, our results make strides to reconcile the association of a non-ion channel protein with epilepsy by suggesting a possible role for EFHC-1 in modulating ion channels to fine-tune neuronal excitation/transmission both at the cilium, the site for sensory input (mechanosensation), and at the synapse, the site for sensory output that regulates the amount of neurotransmitter released.

Previous work has identified EFHC1 at both ciliary (*Conte et al., 2009*; *Ikeda et al., 2003*; *Ikeda et al., 2005*; *Suzuki et al., 2008*; *Suzuki et al., 2009*) and synaptic (*Rossetto et al., 2011*) regions, yet proposed functions at either location have only been considered separately. Our work

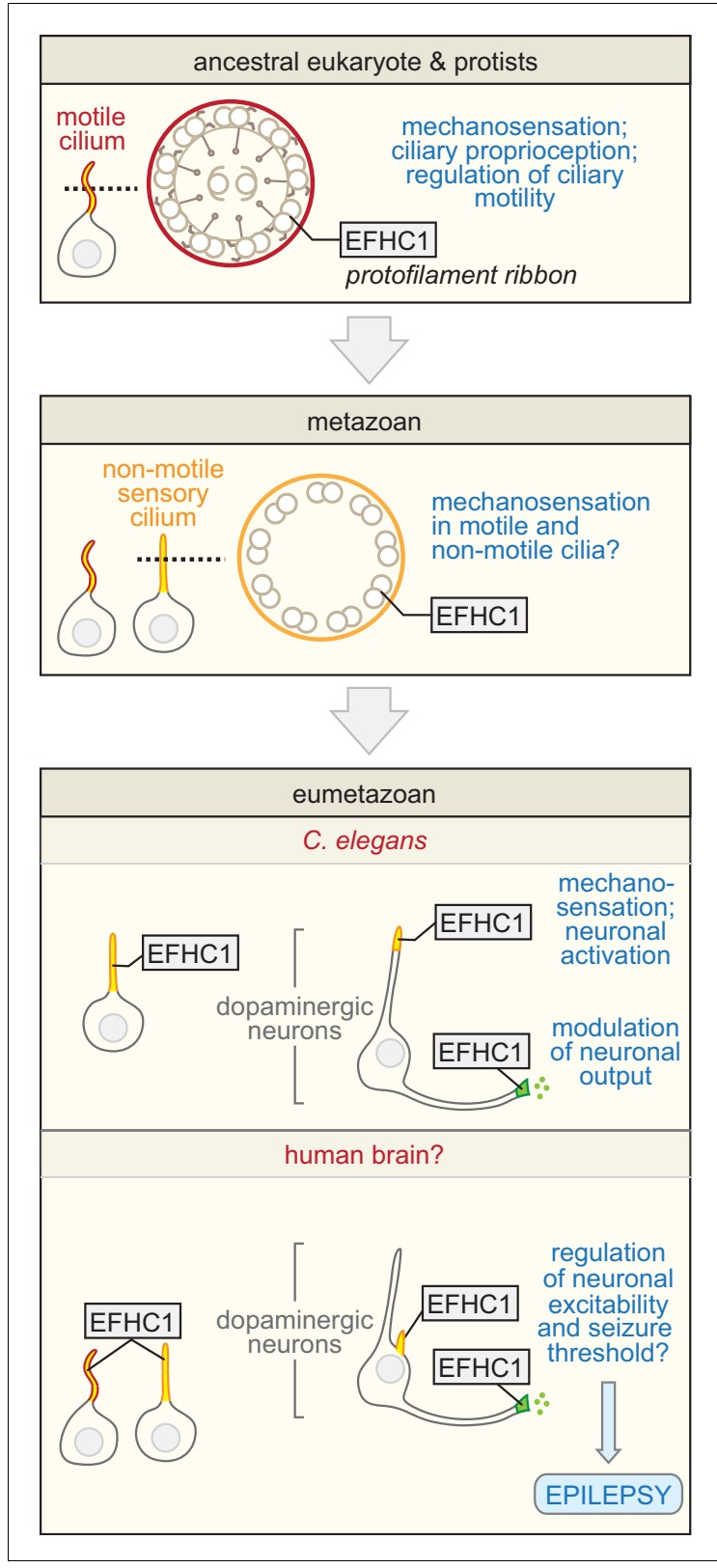

**Figure 5.** Proposed evolutionary model for the exaptation of EFHC1 motility-associated ciliary functions in non-motile cilia and synapses. The role of EFHC1 in extant protists (and probably the ancestral eukaryote) is likely to regulate ciliary motility (*Ikeda et al., 2003*), potentially by acting as a mechanosensor together with a TRP family (or other) calcium channel. In metazoans, this function was retained for motile cilia (*Suzuki et al., 2009*), and

*Figure 5 continued on next page*

*Figure 5 continued*

adapted for mechanosensation and potentially other signaling roles in non-motile cilia. EFHC1 evolved in *C. elegans*, and potentially other neuron-bearing eumetazoans, to also regulate synaptic transmission. In *C. elegans*, the combined functions for EFHC-1 at cilia (mechanosensory input) and at the synapse (output) help to regulate neuronal excitability and dopamine signaling, both of which may be relevant to epilepsy in human patients with mutations in *EFHC1*. Motile and non-motile cilia are shown in red and orange, and synapses in green.

DOI: https://doi.org/10.7554/eLife.37271.017

in *C. elegans* provides a unique perspective given that EFHC-1-containing cells harbor both ciliary and synaptic regions, thus raising the intriguing possibility of differential trafficking to these two distinct locations within single cells. We show that EFHC-1's ciliary localization is independent of UNC-101 and synaptic localization is independent of UNC-104, indicating that EFHC-1's unique localization pattern does not rely on these known active transport mechanisms. While it is possible that EFHC-1's localization pattern is dictated by other transport mechanisms, it could also result from diffusion of EFHC-1 throughout cells, followed by ciliary/synaptic retention through interactions with cytoskeletal arrangements, or specific proteins, unique to each location that enable scaffolding (*e.g.*, microtubule-associated proteins for ciliary localization [*Ikeda et al., 2005*] and actin-associated proteins for synaptic localization [*Torres and Inestrosa, 2018*]). These possibilities warrant further attention to help better understand EFHC-1's mechanism of action at each site.

By identifying an epilepsy-related protein that directly modulates dopamine signaling in *C. elegans*, our work helps to strengthen the potential link between dopamine signaling and epilepsy, where evidence is accumulating to suggest that inappropriate regulation of dopamine signaling may contribute to various forms of epilepsy (*Bozzi and Borrelli, 2013*). A role for dopamine is particularly relevant in JME because patients also display several deficits in cognitive functions influenced by dopamine, such as behavioral disturbances, social integration problems, impaired executive functions, reduced cognitive speed and impaired working memory (*Bailet and Turk, 2000*; *Devinsky et al., 1997*; *Janz, 1985*; *Pascalicchio et al., 2007*; *Swartz et al., 1996*). Interrogation of the dopaminergic system in JME patients has revealed that dopamine signaling is indeed dysregulated in JME patients (*Ciumas et al., 2008*; *Ciumas et al., 2010*; *Odano et al., 2012*). In combination with our work, this suggests that dopamine may play a role in the onset of seizures and/or cognitive impairments associated with JME.

In conclusion, we propose that our discovery of a dual ciliary (sensory input) - synapse (neuronal output) mechanism that regulates neuronal activity may not only apply to *C. elegans* dopaminergic neurons, but could be relevant to mammalian brain neurons as well, which have non-motile cilia. Indeed, dopamine receptors (DRD1, 2, 5) themselves are found on cilia, as are numerous other receptors, including for serotonin, Melanin-concentrating hormone, somatostatin, galanin, and neuropeptide (FF/YY) (*Hilgendorf et al., 2016*). In such neurons (and others), cilia may modulate neuronal excitation thresholds and thus play important roles in collectively influencing behavior, energy homeostasis and other aspects of mammalian physiology. More generally, our work supports the notion that the relationship between sensory structures and synapses (*Kang et al., 2010*) represents an important biological phenomenon that deserves further attention.

## Materials and methods

### Strains

All strains were grown and cultured according to standard procedures on nematode growth medium (NGM) plates with OP50 bacteria as a food source. The *efhc-1(gk424336)* mutant strain with a nonsense mutation approximately halfway through the gene was obtained from the Million Mutation Project (MMP) collection, and the *efhc-1(tm6235)* mutant strain with an in-frame deletion of exons 3–7 was obtained from the Mitani lab. Both mutant strains were backcrossed six times before being used in experiments or crosses (MX1661 *efhc-1(gk424336)* (6X), MX1977 *efhc-1(tm6235)* (6X)). The wild-type strain used in all experiments was the N2 Bristol strain obtained from the Caenorhabditis Genetics Center (CGC). Other strains used were as follows: MX081 *N2;nxEx[Y49A10A.1::gfp +xbx-1::tdTomato +rol-6(su1006)]*; CB1112 *cat-2(e1112)*; RM2702 *dat-1(ok157)*; AQ2044 *ljIs103[Pcat-1::*

*YCD3*]; AQ2815 *trp-4(gk341);ljIs103[Pcat-1::YCD3*]; AQ2829 *ljEx421[Pocr-4::YC3.60 codon optimized*]; VC818 *trp-4(gk341)*; TQ420 *trp-4(sy695);xuEx542[trp-4::yfp +Podr-1::rfp*]; DR86 *daf-19(m86)*; OW47 *unc-2(zf35)*; CX3877 *lin-15(n765);kyIs156[str-1p::odr-10::gfp +lin-15(+)]*.

The following strains were created in this work: MX79 *nxEx279[pefhc-1::efhc-1::gfp +Posm-5:: xbx-1::tdTomato +rol-6(su1006)]*; MX1971 *efhc-1(gk424336)*; *nxEx[pefhc-1::efhc-1 (217–447):: gfp +ccgfp]*; MX2192 *efhc-1(gk424336)*; *nxEx[pefhc-1:: efhc-1 (1–216)::GFP +Posm-5::xbx-1:: tdTomato +ccgfp]*; MX2195 *nxEx[pefhc-1::efhc-1::gfp +pefhc-1::xbx-1::tdtomato +cc::gfp]*; MX2194 *N2;nxEx[snb-1::tdtomato +Y49A10A.1::gfp +cc::gfp]*; MX2132 *efhc-1(gk424336);cat-2(e1112)*; MX2403 *efhc-1(gk424336);ljIs103[Pcat-1::YCD3]*; MX2317 *efhc-1(gk424336);trp-4(sy695);xuEx542 [trp-4::yfp +Podr-1::rfp]*; MX2129 *efhc-1(gk424336);trp-4(gk341)*; MX2316 *efhc-1(gk424336);daf-19 (m86)*; MX1973 *daf-19(m86) II;daf-12(sa204);nxEx[efhc-1::gfp +cc::gfp]*; MX2185 *dpy-5(e907);nxEx [pefhc-1::unc-2(zf35) +cc::gfp +dpy-5(+)]*; MX2341 *efhc-1(gk424336);dpy-5(e907);nxEx[pefhc-1::unc-2 (zf35) +cc::gfp +dpy-5(+)]*; MX2317 *efhc-1(gk424336);trp-4(sy695);xuEx542[trp-4::yfp +Podr-1::rfp]*; MX2137 *efhc-1(gk424336);ljEx421[Pocr-4::YC3.60 codon optimized]*. MX2724 *efhc-1(nx163[efhc-1:: gfp::3xflag])X*; MX2727 *N2;egIs1[pdat-1::gfp];nxEx2714[Pdat-1::elks-1::tdtomato +rol-6(su1006)]*; MX2726 *efhc-1(gk424336);egIs1[pdat-1::gfp];nxEx2714[Pdat-1::elks-1::tdtomato +rol-6(su1006)]*; MX2742 *efhc-1(nx163[efhc-1::gfp::3xflag])X*; *nxEx2714[Pdat-1::elks-1::tdtomato +rol-6(su1006)]*; MX2754 *efhc-1(nx163[efhc-1::gfp::3xflag])X*; *nxEx2754[Pdat-1::elks-1::tdTomato +rol-6(su1006)]*; MX2769 *efhc-1(nx163[efhc-1::gfp::3xflag])X*; *nxEx2769 [Pdat-1::mcherry::rab-3+rol-6(su1006)]*; MX2904 *unc-104(e1265); efhc-1(nx163[efhc-1::gfp::3xFlag])X;nxEx2769[Pdat-1::mCherry::rab-3+rol-6 (su1006)]*; MX2905 *unc-104(e1265); efhc-1(nx163[efhc-1::gfp::3xFlag])X; nxEx2754[Pdat-1::ELKS-1:: tdTomato +rol-6(su1006)]*; MX2908 *unc-101(m1); efhc-1(nx163[efhc-1::gfp::3xFlag])X; nxEx2769 [Pdat-1::mCherry::rab-3+rol-6(su1006)]*; MX2909 *unc-101(m1); lin-15(n765); kyIs156[str-1p::odr-10:: gfp +lin-15(+)]*

## Generation and visualisation of transgenic animals

### Generation of fusion constucts
Both the pefhc-1::XBX-1::tdTomato and pefhc-1::SNB-1::tdTomato constructs were created using 2036 bp of *efhc-1*'s endogenous upstream promoter fused to all the exons/introns of *xbx-1* or *snb-1* and then fused in-frame to tdTomato. The pefhc-1::unc-2(gf) construct was again created using 2036 bp of *efhc-1*'s endogenous upstream promoter fused to the cDNA of *unc-2* with a gain-of-function mutation and a synthetic intron. To construct the presynaptic markers, mcherry::rab-3::unc-10 3' UTR was cloned from a plasmid (kindly provided by the Mizumoto lab), and *elks-1* was amplified from genomic DNA and stitched to the tdTomato::unc-54 3' UTR. Both amplified PCR fragments were stitched to the *dat-1* promoter (~700 bp) to drive expression in the dopaminergic neurons (CEP, ADE and PDE).

### Generation of an efhc-1::gfp knock-in strain
The *unc-119* guide RNA sequence in p*U6::unc-119* sgRNA vector (Addgene #46169) was replaced with the predicted *efhc-1* CRISPR guide RNA sequence (5'- GTACAGTCTAGTGGAGAGA-3'), which is present in exon 10 of *efhc-1* using site-directed mutagenesis technique (http://crispr.mit.edu/) (*Friedland et al., 2013*). The donor template for the C-terminal GFP knock-in was prepared from pDD282 (Addgene #66823) (*Dickinson et al., 2015*). The pDD282 was digested with SpeI and AvrII to remove *ccdB* markers and mixed with upstream and downstream homology (500 ~ 650 bp) arms for *efhc-1*. The donor plasmid was cloned using Gibson Assembly. The GFP donor template plasmid contains T434T silent mutation in PAM site. The mixture (50 ng/μl coel::RFP (Addgene #8938), 50 ng/μl pCAS9 (Addgene #46168), 50 ng/μl *efhc-1* sgRNA, and 50 ng/μl donor template) was injected into 30 adult worms. F1 worms were screened using 250 μg/ml hygromycin solution followed by picking 100% roller F2. A self-excising drug selection cassette (SEC), which contains *sqt-1*, *hs::Cre*, and *hygR*, was removed by heat shock at 34°C for 4 hr. This worm strain was confirmed by Sanger DNA sequencing.

### Visualization of transgenic animals
Fluorescently-tagged proteins were visualized by first immobilizing animals using 0.5 μl of 10 mM levamisole and 0.5 μL of 0.1 μm diameter polystyrene microspheres (Polysciences 00876–15, 2.5%

w/v suspension) on 5% agarose pads. Z-stack images were collected using a spinning disc confocal microscope (WaveFX; Quorum Technologies), a Hammamatsu 9100 EMCCD camera and Volocity software (PerkinElmer). Fluorescence recovery after photobleaching (FRAP) assay was performed using LSM880 laser scanning microscope with Airyscan and ZEN 2.3 software. Thresholded Pearson's correlation coefficients for the colocalization of EFHC-1::GFP and ELKS-1::tdTomato in CEP axons were calculated using Volocity 6.3 software.

## Mating efficiency test

An OP50 culture was grown at 37°C for 16 hr. To make mating plates, 20 μl of the culture was pipetted onto an NGM agar plate and the seeded plates were incubated at 25°C for 20 hr. One heat-shock generated homozygous L4 stage male (wild-type or mutant) and one L4 stage wild-type hermaphrodite were placed onto the bacterial lawn of the mating plate. All mating plates were incubated at 20°C for 4 days. The plates containing male progeny were counted as successful matings.

## Behavioral analyses

All behavioral analyses were conducted on at least two separate days to account for day-to-day differences.

### Tap habituation

Worms were age synchronized by allowing five gravid adults to lay eggs for 3–4 hr on individual NGM plates seeded with *E. coli* OP50 bacteria and testing the worms after 4 days. At least one hour before testing, 20–30 worms were transferred to fresh NGM plates, with or without food. Plates with food were seeded with 50 μL of liquid culture OP50 bacteria in lysogeny broth (LB) 24 hr beforehand, while plates without food were seeded with 50 μL LB. Plates with worms were then placed into the Multi-Worm Tracker (MWT), and after a 300 or 600 s acclimatization period, 30 taps were administered to the side of the plate using a solenoid tapper at 10 s interstimulus intervals (ISI) [31]. The script used for tap habituation analysis is available at *Loucks (2018)* (copy archived at https://github.com/elifesciences-publications/dopamine_habituation). All experiments were done at 20°C, and in most experiments strains were cultured at 20°C. In the habituation experiment involving *daf-19(m86)*, all strains were cultured at 15°C for four days then moved to 20°C for two days to enrich for non-dauer worms in *daf-19* and *daf-19;efhc-1* worms. The script for analysis is available at: https://github.com/cloucks/dopamine_habituation). Experiments from individual days were only included if at least 80% of wild-type animals responded to the initial tap, indicating that laboratory conditions (temperature, age and humidity) were appropriate for the subsequent analyses. Given that the MWT does not track every animal on a plate at a given time, minimum sample sizes for a single tap in a given experiment are presented in *Supplementary file 1*.

### Swimming-induced paralysis (SWIP)

Worms were age synchronized by allowing 10 gravid adults to lay eggs for 3 hr on individual NGM plates seeded with OP50 bacteria. L4 worms were tested after 2 days at 20°C. Two different SWIP assays were performed: one manual and one automated. For both assays, ten L4 animals were placed in 30 μL of tap water in a single well of a Pyrex Spot Plate (Fisher). For the manual assay, animals were assessed for paralysis at 10 min, defined as no C-shaped bends for at least 5 s. For the automated assay, the plate was analysed using the Multi-worm tracker (MWT) to assess thrashing frequency of animals over 11 min (script available at: https://github.com/cloucks/dopamine_swimming).

### Basal slowing response

Worms were age synchronized by allowing five gravid adults to lay eggs for 3–4 hr on individual NGM plates seeded with OP50 bacteria and testing the worms after 3 days. On the day of testing, worms were washed off their individual plates with M9 buffer and transferred to individual NGM with or without food. Plates with food were seeded with 100 μL HB101 24 hr beforehand, while plates without food were seeded with 100 μL LB. Plates with worms were then placed into the Multi-Worm Tracker (MWT) (*Swierczek et al., 2011*) for 600 s. Body bends were counted for animals persisting from 60 to 120 s.

## Suppression of foraging

Suppression of foraging was performed as described previously (*Kindt et al., 2007b*; *Loucks et al., 2016*).

## Single worm tracking behavioral assessment

Single-worm tracking was performed as described previously (*Loucks et al., 2016*; *Yemini et al., 2013*).

## Calcium imaging of CEP and OLQ neurons

Calcium imaging of nose touch stimulation of glued animals was performed using identical protocols to our previous work (*Loucks et al., 2016*), which was essentially as described previously, using a 1 s 'press' or 3.7 s 'buzz' stimulus (*Kindt et al., 2007a*; *Kindt et al., 2007b*). Images were recorded at 10 Hz using an iXon EM camera (Andor Technology), captured using IQ1.9 software (Andor Technology) and analysed using a Matlab (MathWorks) program custom written by Ithai Rabinowitch. A rectangular region of interest (ROI) was drawn around the cell body and for every frame the ROI was shifted according to the new position of the centre of mass. Fluorescence intensity, F, was computed as the difference between the sum of pixel intensities and the faintest 10% pixels (background) within the ROI. Fluorescence ratio $R = F_{yellow}/F_{cyan}$ (after correcting for bleed through) was used for computing ratio change, expressed as a percentage of $R_0$ (the average R within the first 3 s of recording) (*Rabinowitch et al., 2013*).

## Transmission electron microscopy (TEM)

Day one adult worms were fixed (glutaraldehyde), embedded, serial sectioned and imaged as previously described (*Sanders et al., 2015*). The one exception is the wild type image in *Figure 3C*, which was obtained from sections of adult day one worms fixed and embedded using high pressure freezing (Leica EMPACT2) and freeze substitution (FS), as previously described (*Cohen et al., 2008*). The FS step was performed using a solid metal block placed at −75°C for 48 hr, followed by −20°C for 24 hr, and then room temperature for 3 hr.

## Statistical tests

Logistic regression was used for the statistical analyses performed because the dependent variables (proportion reversing to tap, proportion swimming and proportion responding) are not continuous but instead represent binary outcomes (e.g. reversing or not reversing). Futhermore, our data meet the assumptions required for logistic regression, including the independence of sample outcomes, an absence of strongly influential outliers and a sufficient number of recorded events (*Stoltzfus, 2011*). Sample sizes for each behavioral assay were chosen to be either equal to or greater than sample sizes reported in the literature that were sufficient to detect biologically relevant differences.

## Acknowledgments

Some of the nematode strains used in this work were provided by the *Caenorhabditis* Genetics Center (funded by the National Institutes of Health National Center for Research Resources), the Mitani lab and the Million Mutation Project (MMP) collection. We would also like to thank Dr. Mark Alkema for the *unc-2(gf)* strain and associated plasmid, Dr. Shawn Xu for the strain expressing TRP-4::YFP and Dr. Kota Mizumoto for the plasmid carrying *mcherry::rab-3*. Finally, we would like to thank Robyn Branicky for the AQ2044 strain, Ithai Rabinowitch for the AQ2815 and AQ2829 strains and Marios Chatzigeorgiou for the AQ2815 strain.

## Additional information

### Funding

| Funder | Grant reference number | Author |
| --- | --- | --- |
| Canadian Institutes of Health Research | MOP82870 | Michel R Leroux |

| | | |
|---|---|---|
| Michael Smith Foundation for Health Research | Senior Scholar Award | Michel R Leroux |
| Natural Sciences and Engineering Research Council of Canada | Discover Grant 122216-2013 | Catharine H Rankin |
| Science Foundation Ireland | 11/PI/1037 | Oliver E Blacque |
| Medical Research Council | MC_A023_5PB91 | William Schafer |
| Wellcome | WT103784MA | William Schafer |
| Canadian Institutes of Health Research | MOP130287 | Catharine H Rankin |
| Canadian Institutes of Health Research | Frederick Banting and Charles Best Canada Graduate Scholarship | Catrina M Loucks |
| Natural Sciences and Engineering Research Council of Canada | Vanier Canada Graduate Scholarship | Kwangjin Park |

The funders had no role in study design, data collection and interpretation, or the decision to submit the work for publication.

### Author contributions

Catrina M Loucks, Conceptualization, Software, Formal analysis, Investigation, Visualization, Methodology, Writing—original draft, Project administration, Writing—review and editing; Kwangjin Park, Formal analysis, Investigation, Visualization, Writing—review and editing, Performed CRISPR/Cas9 experiments and associated imaging; Denise S Walker, Formal analysis, Investigation, Visualization, Writing—review and editing, Performed calcium imaging experiments; Andrea H McEwan, Investigation, Writing—review and editing, Collected data for the preliminary tap habituation experiments; Tiffany A Timbers, Conceptualization, Methodology, Writing—review and editing, Provided helpful discussions for the project; Evan L Ardiel, Software, Investigation, Writing—review and editing, Developed the preliminary analysis for the automated SWIP assay; Laura J Grundy, Investigation, Writing—review and editing, Performed single-worm tracking experiments; Chunmei Li, Investigation, Visualization, Writing—review and editing, Performed CRISPR/Cas9 experiments and associated imaging; Jacque-Lynne Johnson, Investigation, Writing—review and editing, Injected several constructs to create transgenic strains; Julie Kennedy, Investigation, Writing—review and editing, Performed the transmission electron microscopy analysis; Oliver E Blacque, William Schafer, Catharine H Rankin, Conceptualization, Supervision, Writing—review and editing; Michel R Leroux, Conceptualization, Supervision, Funding acquisition, Writing—original draft, Project administration, Writing—review and editing

### Author ORCIDs

Michel R Leroux [iD] http://orcid.org/0000-0003-0788-9298

### Decision letter and Author response

Decision letter https://doi.org/10.7554/eLife.37271.024
Author response https://doi.org/10.7554/eLife.37271.025

## Additional files

### Supplementary files

• Supplementary file 1. Sample sizes (n) for tap habituation experiments.
DOI: https://doi.org/10.7554/eLife.37271.018

• Supplementary file 2. p values for statistical comparisons.
DOI: https://doi.org/10.7554/eLife.37271.019

• Transparent reporting form
DOI: https://doi.org/10.7554/eLife.37271.020

## Data availability

All data (including data generated during the revision process) have been deposited in Dryad, 10.5061/dryad.bb46h5s.

The following dataset was generated:

| Author(s) | Year | Dataset title | Dataset URL | Database and Identifier |
|---|---|---|---|---|
| Loucks CM, Walker DS, Park K, McEwan AH, Timbers TA, Ardiel EL, Grundy LJ | 2019 | Data from: EFHC1, implicated in juvenile myoclonic epilepsy, functions at the cilium and synapse to modulate dopamine signaling | https://doi.org/10.5061/dryad.bb46h5s | Dryad Digital Repository, 10.5061/dryad.bb46h5s |

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
