## [Decision Letter]

Thank you for submitting your article "EFHC1, implicated in juvenile myoclonic epilepsy, functions at the cilium and synapse to modulate dopamine signaling" for consideration by *eLife*. Your article has been reviewed by Gary Westbrook as the Senior Editor, a Reviewing Editor, and two reviewers. The reviewers have opted to remain anonymous. The reviewers have discussed the reviews with one another and the Reviewing Editor has drafted this decision to help you prepare a revised submission.

Summary:

The reviewers appreciated the importance of the overall findings but have a number of requests for improvement. The most major issue, crystallized during discussions, is the assessment of active transport to synapses and cilia. For the active synapse transport analysis, we recommend that you use the unc-104 mutant, a commonly used tool to confirm synaptic transport. For cilia transport, we recommend standard live visualization of transport (e.g. in combination with FRAP).

*Reviewer #1:*

This is an interesting paper describing the role of the protein EFHC1 in mechanosensation in *C. elegans*. The authors show that the protein localizes to dopaminergic neurons in the head of the animal at both the sensory nonmotile cilium and the synapse. The authors describe defects in the EFHC1 mutant that appear to mimic increase dopaminergic transmission. They suggest that this protein acts independently of the main cilium TRP channel that transduces mechanical stimuli, and may act together with a synaptic calcium channel, UNC2.

EFHC1 mutations are associated with human disease, and the biology of cilia is generally not very well understood. Since these organelles are prevalent, any additional studies are of interest. Here, the coupling between cilium and synapses is intriguing. In that respect, the authors should cite highly relevant previous work in this area (e.g. Shaham, 2010).

The paper is clearly written and the conclusions drawn generally justified. The connection to dopamine signaling is convincing. The paper does not quite address what EFHC1 is doing, however. Some obvious possibilities, such as localization of the TRP channel are ruled out. The UNC2 studies are interesting, but physical association and a clear functional relationship are not established, as the authors use an activated UNC2 allele which may have activities not normally seen in the wild type.

In sum- the data in the paper are solid and convincing. However, a mechanism for EFHC1 function is not established.

*Reviewer #2:*

The manuscript by Loucks et al. characterizes the *C. elegans* ortholog of the ciliary protein EFHC1 that has been linked to Juvenile Myoclonic Epilepsy. Based on its axonal localization, authors argue that EFHC-1 act at this site to modulate dopamine signaling and conclude "Our findings unveil a previously undescribed dual-regulation of neuronal excitability at sites of neuronal sensory input (cilium) and neuronal output (synapse)." While it is clear that EFHC-1 modulates harsh touch habituation (as measured by behavioral assays and calcium imaging), I do not think authors data support the assertion that EFHC-1 functions at the synapse and cilium. Figure 1 shows EFHC-1::GFP localization throughout neurons, including cilia, dendrites, and axons and EFHC-1 colocalization with RAB-3 and ELKS^-1^ is not convincing (Supplemental figure 2). However, authors do have the tools to address my major concern. Is EFHC-1 actively transported in dendrites and axons to its site of action? Active transport to axons would be convincing – diffusion would not.

Does EFHC-1 ciliary vs synaptic localization rely on the same elements? Authors show that EFHC-1(217-417), a construct containing only the second DM10 domain, still localized to cilia. How do EFHC-1 (1-216) and EFHC-1(217-417) look in the rest of the neuron?

Authors show that efhc-1 CEP cilia (by TEM) and synapses (by ELKS^-1^::tdTomato) look normal. The former is convincing, the latter not. How does neuronal architecture look in the efhc-1 mutant (TEM not required, a soluble GFP showing that axonal architecture would suffice).

Figure 3C is really striking. The tubule associated material appears to be surrounded by singlet microtubules. Is there a stereotypical pattern in WT and is this changed at all in efhc-1 mutant? How many cilia were analyzed in WT and efhc-1 animals?

Figure 3D. Authors show TRP-4 localization in WT and efhc-1 cilia and conclude there is no difference. To my eyes, the efhc-1 cilia look longer and TRP-4 more distally located. Please quantitate TRP-4 distribution.

Please provide information on the nature of the efhc-1 alleles.

Figure 1B and Figure 1—figure supplement 1: In the male tail, what is the circular structure outlined by efhc-1::GFP?

Do efhc-1 males display any ray-mediated mating defects?

Figure 2—figure supplement 1: why show amphid channel ultrastructure of efhc-1 mutants if the gene is not expressed in these ciliated cell types? It would be more informative to more carefully analyze CEP and OLQ ultrastructure in efhc-1 animals.

[Editors’ note: the decision after re-review follows.]

Thank you for submitting your work entitled "EFHC1, implicated in juvenile myoclonic epilepsy, functions at the cilium and synapse to modulate dopamine signaling" for consideration by *eLife*. Your revised article has been re-reviewed by two peer reviewers, and the evaluation has been overseen by a Reviewing Editor and a Senior Editor. Our decision has been reached after consultation between the reviewers that is summarized below. Based on these discussions and the individual reviews below, we regret to inform you that your work will not be considered further for publication in *eLife*.

Summary of reviewer discussion:

The authors have done the suggested experiments, but we still don't know if EFHC1 is acting in the axon. The negative results are interesting, suggesting that targeting of EFHC1 occurs by some novel mechanism. However, the bottom line is a bit unsatisfying, as we still don't know details about the mechanism of localization or function. The colocalization of EFHC-1 and ELKS^-1^ at active zones (which the authors say is better imaged in new Figure 1—figure supplement 5BFigure) is not convincing without seeing yellow puncta and without any quantification. This also puts in question the argument that these active zone proteins don't require UNC-104 for location because the data do not provide convincing evidence that EFHC-1 is an active zone protein. Overall, EFHC1 seems to be in the vicinity of synapses, but whether this means anything functionally has not been determined, nor has it been determined the mechanism of differential localization. Without more evidence along these lines, we think the manuscript would be better suited to another journal. We hope that the reviewer comments will be helpful to you.

[Editors’ note: the editors agreed to consider a further resubmission, in which the authors discuss how they interpret the *unc-10/unc-104* results and also provide the requested quantification.]

---

## [Author Response]

Summary:The reviewers appreciated the importance of the overall findings but have a number of requests for improvement. The most major issue, crystallized during discussions, is the assessment of active transport to synapses and cilia. For the active synapse transport analysis, we recommend that you use the unc-104 mutant, a commonly used tool to confirm synaptic transport. For cilia transport, we recommend standard live visualization of transport (e.g. in combination with FRAP).

We very much appreciate having been given the opportunity to submit a significantly revised manuscript which includes additional experimental data. In particular, we specifically address with new experiments the question of how EFHC-1 may be transported to synapses, cilia, and within the ciliary organelle.

Below, we present a point-by-point explanation of how we address these and other comments/concerns presented by the referees, some of which involved text changes, improved figures, and added experimental data.

Reviewer #1:This is an interesting paper describing the role of the protein EFHC1 in mechanosensation in C. elegans. The authors show that the protein localizes to dopaminergic neurons in the head of the animal at both the sensory nonmotile cilium and the synapse. The authors describe defects in the EFHC1 mutant that appear to mimic increase dopaminergic transmission. They suggest that this protein acts independently of the main cilium TRP channel that transduces mechanical stimuli, and may act together with a synaptic calcium channel, UNC2.EFHC1 mutations are associated with human disease, and the biology of cilia is generally not very well understood. Since these organelles are prevalent, any additional studies are of interest. Here, the coupling between cilium and synapses is intriguing. In that respect, the authors should cite highly relevant previous work in this area (e.g. Shaham, 2010).

We thank the reviewer for their overall appreciation of our work and findings. We agree that including a citation to a review that discusses the functional relationship between sensory organs (e.g., cilia) and synapses would be very useful. We now include a sentence explaining this together with the suggested Shaham citation (“Chemosensory organs as models of neuronal synapses”) into the Discussion section.

The paper is clearly written and the conclusions drawn generally justified. The connection to dopamine signaling is convincing. The paper does not quite address what EFHC1 is doing, however. Some obvious possibilities, such as localization of the TRP channel are ruled out. The UNC2 studies are interesting, but physical association and a clear functional relationship are not established, as the authors use an activated UNC2 allele which may have activities not normally seen in the wild type.In sum- the data in the paper are solid and convincing. However, a mechanism for EFHC1 function is not established.

We appreciate the reviewer’s concern that this study does not clearly address the central question of what the function of EFHC1 is at the molecular level. Our manuscript mentions the reported interaction between EFHC1 and the R-type voltage-gated calcium channel. Two complementary approaches based on EFHC1 constructs instead of nonspecific antibodies were used to support the interaction of EFHC1 and Cav2.3

(electrophysiological analyses and co-immunoprecipitation; Suzuki et al., 2003). Our own efforts to carry out pull-down interaction studies in the worm, which is known for not being readily amenable to biochemical analysis, have not been successful. Part of the reason is that cilium-associated proteins such as EFHC1 are only expressed, typically at (very) low levels, in a small subset of cells within the organism (at maximum, 60 of nearly 1000 cells, but lower in this case).

As such, we have resorted to genetic analyses to infer functional interactions between EFHC-1 and UNC-2. We now include truncation analyses of EFHC-1 (Figure 1D) which provide mechanistic insights for how the C-terminal region is sufficient for ciliary localisation; notably, this region contains only one of two poorly-studied DM10 domains found in the protein. While additional mechanistic insights into EFHC1 will be necessary (and this is mentioned in our Discussion section), we hope that on the whole, our discovery of a link between EFHC-1 and dopamine signalling will be of great interest to researchers interested in this protein, cilia, dopamine signalling, epilepsies and synapses.

Reviewer #2:The manuscript by Loucks et al. characterizes the C. elegans ortholog of the ciliary protein EFHC1 that has been linked to Juvenile Myoclonic Epilepsy. Based on its axonal localization, authors argue that EFHC-1 act at this site to modulate dopamine signaling and conclude "Our findings unveil a previously undescribed dual-regulation of neuronal excitability at sites of neuronal sensory input (cilium) and neuronal output (synapse)." While it is clear that EFHC-1 modulates harsh touch habituation (as measured by behavioral assays and calcium imaging), I do not think authors data support the assertion that EFHC-1 functions at the synapse and cilium. Figure 1 shows EFHC-1::GFP localization throughout neurons, including cilia, dendrites, and axons and EFHC-1 colocalization with RAB-3 and ELKS^-1^ is not convincing (Supplemental figure 2). However, authors do have the tools to address my major concern. Is EFHC-1 actively transported in dendrites and axons to its site of action? Active transport to axons would be convincing – diffusion would not.

Although overexpressed GFP transgenes show localisation patterns consistent with that of endogenous protein localisation, it is possible that such localisation may not accurately represent the endogenous localisation of EFHC-1. To address this possibility, we tagged the endogenous *efhc-1* locus with GFP using the CRISPR genome editing tool and determined the localisation of the EFHC-1 expressed at physiological levels. *efhc-1::gfp* is expressed in OLQ, CEP and ADE neurons, and the EFHC-1 protein localises to the OLQ, CEP and ADE cilia. Given that both the overexpressed EFHC-1::GFP construct and the endogenously GFP-tagged EFHC-1 localise specifically to cilia as well as synaptic regions, we feel that this provides strong evidence for the function of EFHC-1 at these two subcellular regions. Consistent with this, EFHC1 is a known cilium-associated protein (*e.g.*, in *Chlamydomonas*, see Ikeda et al., 2005), and in a *Drosophila* study, was also found to localise to the synaptic region (Rossetto et al., 2011).

To further support our assertion that EFHC-1 functions at the cilium and the synapse as recommended by the reviewer, we performed fluorescence recovery after photobleaching (FRAP) in the EFHC-1::GFP overexpression strain and the endogenously GFP-tagged EFHC-1 strain to examine whether EFHC-1 proteins can freely diffuse inside cilia. GFPtagged ARL-13 shows rapid recovery in bleached cilia, indicative of its ciliary localisation being dependent on diffusion (Cevik et al., 2013). In contrast, when we photobleached a portion of either CEP or OLQ cilia, we could not observe a similar rapid recovery within 30 seconds, indicating that EFHC-1’s localisation to cilia is mostly static (Figure 1—figure supplement 4A-B). Moreover, we generated kymographs to observe potential associations between intraflagellar transport (IFT) and EFHC-1, a scenario that is observed for several ciliary proteins. Unlike IFT proteins, we could not observe any clear (anterograde/retrograde) IFT tracks, suggesting that ciliary-localised EFHC-1 proteins are not associated with IFT (Figure 1—figure supplement 4C). Lastly, to determine whether dendritic transport is essential for ciliary localization of EFHC-1, we investigated the ciliary localisation of a control protein, the odorant receptor, ODR-10, that is dependent on dendritic transport mediated by UNC-101 (Dwyer et al., 2001). GFPtagged ODR-10, which is expressed in AWB neurons, mainly accumulates in cilia of wild-type animals, but diffuses throughout the whole AWB neurons in *unc-101* mutants. In contrast, we found that the ciliary localization of EFHC-1 was unchanged in *unc-101* mutants, suggesting that UNC-101 is dispensable for the OLQ and CEP ciliary localisation of EFHC-1 (Figure 1—figure supplement 4D).

We agree that the quality of the EFHC-1 colocalisation images with RAB-3 and ELKS^-1^ needed to be improved to adequately infer the suggested colocalisations. We have replaced these images with ones that are much clearer (Figure 1—figure supplement 5). We also thank the reviewers for suggesting the important experiment of directly assessing active transport of EFHC-1 to synapses using the *unc-104* mutant. Synaptic vesicle proteins like RAB-3 are not transported to synaptic regions from cell bodies in the *unc104 (e1265*) mutant (Nonet et al., 1997). However, we discovered that the axonal localisation of EFHC-1 is UNC-104 independent. Importantly, such a localisation pattern that is independent of UNC-104 transport is also seen for active zone proteins such as ELKS^-1^ (Deken et al., 2005), providing further support that EFHC-1 localises to presynaptic active zones. We include this data in Figure 1—figure supplement 5.

Does EFHC-1 ciliary vs synaptic localization rely on the same elements? Authors show that EFHC-1(217-417), a construct containing only the second DM10 domain, still localized to cilia. How do EFHC-1 (1-216) and EFHC-1(217-417) look in the rest of the neuron?

We have compared the localisation pattern of EFHC-1(1-216) and EFHC-1(217-417) in the rest of the neurons. We found that EFHC-1(217-417) predominantly localizes to the cilia of dopaminergic (CEP, ADE, and PDE) as well as glutamatergic (OLQ) neurons, but is weakly present in the rest of the neurons (dendrites, cell bodies, axons). However, EFHC-1(1-216) variant diffuses throughout the neurons but is weakly present at the cilium (Figure 1—figure supplement 3). For these constructs, the differences between synaptic localisations are less apparent, preventing any conclusion that either truncation (each containing one DM10 domain) is responsible for facilitating synaptic localization of EFHC-1.

Authors show that efhc-1 CEP cilia (by TEM) and synapses (by ELKS^-1^::tdTomato) look normal. The former is convincing, the latter not. How does neuronal architecture look in the efhc-1 mutant (TEM not required, a soluble GFP showing that axonal architecture would suffice).

We thank the reviewer for this excellent suggestion. To address this question, we expressed *Pdat-1::gfp* to specifically mark dopaminergic neurons in wild-type and *efhc-1 (gk424336*) mutant strains. We discovered that the axonal architecture of the *efhc-1* mutants is indistinguishable from that of wild-type animals. We have replaced the previous image in Figure 4—figure supplement 1.

Figure 3C is really striking. The tubule associated material appears to be surrounded by singlet microtubules. Is there a stereotypical pattern in WT and is this changed at all in efhc-1 mutant? How many cilia were analyzed in WT and efhc-1 animals?

The TAM in the CEP cilium appears similarly organised in wild-type animals and *efhc-1* mutants; typically 7-8 individual TAM densities are visible, each surrounded by multiple singlet microtubules. We counted microtubule numbers in three wild-type cilia (33, 27, 45) and two *efhc-1* mutant cilia (31, 40). On average, we see 35/36 singlet microtubules per CEP cilium in each case indicating no gross defect in CEP microtubule number.

Figure 3D. Authors show TRP-4 localization in WT and efhc-1 cilia and conclude there is no difference. To my eyes, the efhc-1 cilia look longer and TRP-4 more distally located. Please quantitate TRP-4 distribution.

We have measured the length of CEP and OLQ cilia in *trp-4* and *trp-4;efhc-1* mutants using the TRP-4::YFP marker. We found that ciliary length was not changed, and added these data to Figure 3D. We also replaced the previous images with ones that are of better quality, and are more representative, of the localisation of TRP-4::YFP to these cilia in these mutants.

Please provide information on the nature of the efhc-1 alleles.

The *efhc-1* alleles used for these images are the *gk424336* allele with a nonsense mutation (L222Amber), and the *tm6235* allele with an in-frame deletion of exons 3–7 (Loucks et al., 2016). These details have been added to the Materials and methods section.

Figure 1B and Figure 1—figure supplement 1: In the male tail, what is the circular structure outlined by efhc-1::GFP?

We are uncertain as to what the circular structure is, but it is not consistently observed across different microscopy images. We chose this particular image for the manuscript because it shows a clear localisation pattern for the EFHC-1 protein distal to the dendrites of dopaminergic neurons within the male tail.

Do efhc-1 males display any ray-mediated mating defects?

This is an excellent question, given the role of these structures in male reproductive behavior. To address this, we have examined the mating efficiency of *efhc-1* mutants that show increased dopamine signalling according to both the tap habituation and the swimming-induced paralysis (SWIP) assays. While the dopamine-deficient mutant *cat-2 (e1112*) males show mating deficits (Correa et al., 2012), *efhc-1 (tm6235* and *gk424336*) mutant males have similar mating potency to wild-type animals (Figure 2—figure supplement 1B). This suggests that enhanced dopamine signalling does not interfere with male mating.

Figure 2—figure supplement 1: why show amphid channel ultrastructure of efhc-1 mutants if the gene is not expressed in these ciliated cell types? It would be more informative to more carefully analyze CEP and OLQ ultrastructure in efhc-1 animals.

The OLQ cilia of *efhc-1* mutants display normal ultrastructure; distal segments display the characteristic square arrangement of 4 connected electron dense doublet microtubules, followed by a middle segment of variable doublet microtubule numbers, and a proximal transition zone of 9 doublet microtubules with clearly evident Y-links. As mentioned above, the CEP cilia of *efhc-1* mutants also show normal ultrastructure with similar arrangements of singlet microtubules surrounded by tubule-associated material. We chose to also look at the ciliary ultrastructure of cilia to which EFHC-1::GFP was not observed to account for the possibility that EFHC-1 may be present within these cilia at levels below detection capabilities. It was thus not surprising that no ciliary ultrastructure defects were seen in other ciliated cell types. We thank the reviewer for this question and added text to help explain this in the manuscript.

[Editors’ note: the author response to the re-review process follows.]

[…] The authors have done the suggested experiments, but we still don't know if EFHC1 is acting in the axon. The negative results are interesting, suggesting that targeting of EFHC1 occurs by some novel mechanism. However, the bottom line is a bit unsatisfying, as we still don't know details about the mechanism of localization or function. The colocalization of EFHC-1 and ELKS^-1^ at active zones (which the authors say is better imaged in new Figure 1—figure supplement 5B) is not convincing without seeing yellow puncta and without any quantification. This also puts in question the argument that these active zone proteins don't require UNC-104 for location because the data do not provide convincing evidence that EFHC-1 is an active zone protein. Overall, EFHC1 seems to be in the vicinity of synapses, but whether this means anything functionally has not been determined, nor has it been determined the mechanism of differential localization. Without more evidence along these lines, we think the manuscript would be better suited to another journal. We hope that the reviewer comments will be helpful to you.

We would like to thank the reviewers for their interest in how EFHC-1 is targeted to the presynaptic regions, and how this relates to its function.

First, we agree that the previous colocalization images of EFHC-1 and ELKS^-1^ at synaptic regions were not sufficiently convincing. The orientation of images was originally chosen to best visualize cell bodies and axons, but we now include colocalization images in two different orientations, where the second (flipped 180° through the z axis) shows yellow puncta more clearly. These data are in the revised Figure 1—figure supplement 5B (last frame represents the added image).

These colocalization images for both wild-type and *unc-104* animals show partial colocalization in axons, as quantified using the Pearson Correlation Coefficient (PCC). We find that the average PCC of EFHC-1::GFP and ELKS^-1^::tdTomato is 0.2541 in wild-type animals and 0.243 in *unc-104* animals. The quantitation results are found in Figure 1—figure supplement 5C.

Since a value of 1 indicates perfect colocalization, our results indicate only partial colocalization between EFHC-1 and ELKS^-1^, an established active zone protein. This colocalization is not significantly different upon the loss of UNC-104, the kinesin motor implicated in the axonal transport of vesicles to synapses.

Together, our data support the assertion that endogenous EFHC-1 tagged with GFP localizes to synaptic regions in close proximity to the active zone protein, ELKS^-1^, and that the synaptic localization pattern of EFHC-1 does not appear to rely on UNC-104, much like ELKS^-1^ itself: see Nonet et al., 1997; Deken et al., 2005; and Patel et al., 2006.

These results do not allow us to classify EFHC-1 as an active zone protein (as the reviewers pointed out), and we have ensured that our manuscript text only specifies that EFHC-1 localizes in close proximity to active zones. The relevant change within the text of the manuscript is shown below (presynaptic regions is now used instead of presynaptic active zones):

In addition to EFHC-1’s ciliary localization, we find that the EFHC-1::GFP fusion protein is present at presynaptic regions in dopaminergic neurons, partially overlapping with the synaptic vesicle protein, synaptobrevin (SNB-1) (Figure 1E). By tagging endogenous EFHC-1 with GFP, we confirm that EFHC-1 localizes to these presynaptic regions, where it partially overlaps with a second synaptic vesicle protein (RAB-3) (Figure 1—figure supplement 5A). More specifically, EFHC-1 is found to partially colocalize with the active zone protein (ELKS^-1^) (Figure 1—figure supplement 5B-C), where it is ideally positioned to regulate synaptic release in *C. elegans*. Since the presynaptic localization of synaptic vesicle proteins such as RAB-3, but not some active zone proteins like ELKS^-1^, depends on the kinesin-3 motor, UNC-104 (Deken et al., 2005; Nonet et al., 1997), we examined whether EFHC-1 also requires UNC-104 for its synaptic localization in CEP neurons. While RAB-3 localization is abrogated in *unc-104* mutants, accumulating in cell bodies and diffusing to dendrites/axons of CEP neurons, UNC-104 is dispensable for both ELKS^-1^ and EFHC-1 localization (and consequently their partial colocalization pattern) to presynaptic regions (Figure 1—figure supplement 5A-C). The presence of EFHC-1 at both the cilium and presynaptic regions of dopaminergic neurons suggests that it may play important roles in both sensation and regulation of neuronal output.

Notably, the localization (and function) of EFHC-1 at synaptic regions is further supported by our genetic analyses with UNC-2, a protein localized at presynaptic active zones and involved in provoked neurotransmitter release at the synapse. Specifically, we provide evidence, using tap habituation and swimming induced paralysis assays, both of which are modulated by dopamine signaling, that EFHC-1 negatively regulates UNC-2 activity (Figures 4C and Figure 2—figure supplement 2E). Importantly, human EFHC1 has been found to physically interact with the R-type voltage-dependent calcium channel, Ca_v_2.3, which is related to *C. elegans* UNC-2 (Suzuki et al., 2004). There is also consensus in the literature that EFHC1 likely interacts with Ca_v_2.3 to modulate its function (see Suzuki et al., 2004 and Katano et al.,2012). In summary, these data support a role for EFHC1 at synaptic regions, consistent with previously published work.

In conducting the requested experiments to help understand how EFHC-1 is transported to ciliary and synaptic regions in glutamatergic and mechanosensory neurons in *C. elegans*, we show that the localizations of EFHC-1 are independent of either UNC-101 (dendritic transport) or UNC-104 (axonal transport). This indicates that EFHC-1 may be transported to either location viaother transport mechanisms. Furthermore, our results highlight new avenues of research that might help to better understand this epilepsy-associated protein. To further address this in the manuscript, we include a paragraph in the discussion in which we bring attention to the importance of this area for better understanding EFHC-1’s proposed functions. Specifically, we propose that EFHC-1 may differentially localize to ciliary/synaptic regions through diffusion, followed by specific retention at these locations, possibly facilitated by cytoskeletal arrangements and associated proteins unique to each location (microtubule arrangements such as the EFHC-1-containing protofilament ribbon at the cilium, or actin arrangements that act as scaffolds to organize synaptic release machinery close to active zones). The paragraph we added to the Discussion section is shown below:

Previous work has identified EFHC1 at both ciliary (Conte et al., 2009; Ikeda et al., 2003b, 2005; Suzuki et al., 2008, 2009) and synaptic (Rossetto et al., 2011) regions, yet proposed functions at either location have only been considered separately. Our work in *C. elegans* provides a unique perspective given that EFHC1-containing cells harbor both ciliary and synaptic regions, thus raising the intriguing possibility of differential trafficking to these two distinct locations within single cells. We show that EFHC-1’s ciliary localization is independent of UNC-101 and synaptic localization is independent of UNC-104, indicating that EFHC-1’s unique localization pattern does not rely on theseknown active transport mechanisms. While it is possible that EFHC-1’s localization pattern is dictated by other transport mechanisms, it could also result from diffusion of EFHC-1 throughout cells, followed by ciliary/synaptic retention through interactions with cytoskeletal arrangements, or specific proteins, unique to each location that enable scaffolding; for example, microtubule-associated proteins for ciliary localization (Ikeda et al., 2005) and actin-associated proteins for synaptic localization (Torres and Inestrosa, 2018). These possibilities warrant further attention to help better understand EFHC-1’s mechanism of action at each site.

We agree that how EFHC-1 differentially localizes to ciliary and synaptic regions is an interesting and important problem, but our work has also uncovered several important insights into the functions of EFHC-1 in *C. elegans* neurons that may have parallels in the human brain. Firstly, we show that EFHC-1 allows for normal dopamine signaling, suggesting that in its absence dopamine transmission is impaired, a phenomenon that may have consequences for neuronal excitability, leading to epilepsy in the human brain (Bozzi and Borrelli, 2013 and Ciumas et al., 2008).

Secondly, we find that EFHC-1 is required for mechanosensation, another novel finding given that EFHC-1 has previously been associated with ciliary motility and not sensory functions. Given that the loss of EFHC-1 prevents the mechanosensation required for dopamine release and yet also increases dopamine signaling (as measured by several behavioral assays), our work demonstrates that EFHC-1 localizes to and functions at the cilium, as well as downstream (in close proximity to synaptic active zones) to facilitate dopamine signaling.

This dual localization and function of EFHC-1, which regulates neuronal sensory input and output to allow appropriate dopamine release/signaling, represents a novel, central finding of our study. We hope our findings will not only help illuminate the function of EFHC-1, but also provide interesting insights into the connection between dopamine signalling, cilia, the synapse, and epilepsy.